# Tumor immune microenvironment permissive to metastatic progression of ING4-deficient breast cancer

Emily Tsutsumi[1,2], Anne M. Macy[3,4], Janine LoBello[5¤], Karen T. Hastings[3,4], Suwon Kim[1,2]*

1 Department of Basic Medical Sciences, University of Arizona College of Medicine-Phoenix, Phoenix, Arizona, United States of America, 2 Cancer and Cell Biology Division, Translational Genomic Research Institute, Phoenix, Arizona, United States of America, 3 Department of Dermatology, University of Arizona College of Medicine-Phoenix, Phoenix, Arizona, United States of America, 4 Phoenix Veterans Affairs Health Care System, Phoenix, Arizona, United States of America, 5 Cancer Genomics Division, Translational Genomics Research Institute, Phoenix, Arizona, United States of America

¤ Current address: Exact Sciences, Phoenix, Arizona, United States of America
* suwon@arizona.edu

**Data Availability Statement:** All relevant data are within the manuscript.

**Funding:** The work was supported by National Health Institute/National Cancer Institute R03

## Abstract

Deficiencies in the ING4 tumor suppressor are associated with advanced stage tumors and poor patient survival in cancer. ING4 was shown to inhibit NF-kB in several cancers. As NF-kB is a key mediator of immune response, the ING4/NF-kB axis is likely to manifest in tumor-immune modulation but has not been investigated. To characterize the tumor immune microenvironment associated with ING4-deficient tumors, three approaches were employed in this study: First, tissue microarrays composed of 246 primary breast tumors including 97 ING4-deficient tumors were evaluated for the presence of selective immune markers, CD68, CD4, CD8, and PD-1, using immunohistochemical staining. Second, an immune-competent mouse model of ING4-deficient breast cancer was devised utilizing CRISPR-mediated deletion of *Ing4* in a *Tp53* deletion-derived mammary tumor cell line; mammary tumors were evaluated for immune markers using flow cytometry. Lastly, the METABRIC gene expression dataset was evaluated for patient survival related to the immune markers associated with *Ing4*-deleted tumors. The results showed that CD68, CD4, CD8, or PD-1, was not significantly associated with ING4-deficient breast tumors, indicating no enrichment of macrophages, T cells, or exhausted T cell types. In mice, *Ing4*-deleted mammary tumors had a growth rate comparable to *Ing4*-intact tumors but showed increased tumor penetrance and metastasis. Immune marker analyses of *Ing4*-deleted tumors revealed a significant increase in tumor-associated macrophages (Gr-1$^{lo}$CD11b$^+$F4/80$^+$) and a decrease in granzyme B-positive (GzmB+) CD4$^+$ T cells, indicating a suppressive and/or less tumoricidal immune microenvironment. The METABRIC data analyses showed that low expression of *GZMB* was significantly associated with poor patient survival, as was *ING4*-low expression, in the basal subtype of breast cancer. Patients with *GZMB*-low/*ING4*-low tumors had the worst survival outcomes (HR = 2.80, 95% CI 1.36–5.75, p = 0.0004), supportive of the idea that the *GZMB*-low immune environment contributes to ING4-

CA270486 (SK), Valley Research Program P1 #VRP68 (SK, ET), and the Fidelity Charitable Donor-Advised Yoo Family fund grant (SK). The funders had no role in the study design, data collection, analysis, decision to publish, or preparation of the manuscript.

**Competing interests:** The authors have declared that no competing interests exist.

deficient tumor progression. Collectively, the study results demonstrate that ING4-deficient tumors harbor a microenvironment that contributes to immune evasion and metastasis.

## Introduction

The tumor immune microenvironment (TIME) plays a critical role in all stages of tumorigenesis including initiation, progression, and metastasis [1–3]. Studies have shown that tumors developing in the absence of host immune components are more immunogenically diverse than tumors in immune-intact hosts, providing evidence that tumor clones are immunologically eliminated during the early stages of tumorigenesis and that tumors arise from having evaded immune recognition [4,5]. Moreover, the TIME of established tumors exhibit suppressive and/or exhausted immune phenotypes, indicating that tumor-immune modulation toward inactive immunity enables tumor progression [6,7]. Reactivation of immune cells via immune-checkpoint blockade has been shown effective as therapy in several cancers [8,9], further validating that immune cells exert pivotal effects of tumor elimination vs promotion. Accordingly, characterization of the immune cell types and their functional states in the TIME has been the focus of many studies [10,11].

In breast cancer, the histopathologic presence of tumor infiltrating lymphocytes (TILs) has been associated with good prognosis [12–14]. Studies have further characterized that TILs consisting of an increased CD8$^+$/CD4$^+$ T cell ratio and of natural killer (NK) cells correlated with patient survival and therapy response in breast cancer [15–18], suggesting that cytotoxic immunity favors tumor destruction. More recently however, tumor transcriptomics and single cell analyses have revealed complex landscapes of immune cells and various signatures depending on tumor subtypes, stages, and/or therapy [19–21]. Thus, deciphering specific immune compositions of the TIME associated with tumor parameters remains a challenge [22,23]. This is undoubtedly due to the multiple factors that govern tumor-immune interactions, including the heterogeneity of tumor mutations. Indeed, driver mutations in several oncogenes and tumor suppressors have been directly implicated in tumor-immune modulation [24–27]. Further characterization of the TIMEs associated with specific tumor genotypes may provide insight into the mechanisms of tumor-immune modulation and enable therapy strategies targeting immune components.

ING4 is a member of the Inhibitor of Growth (ING) family and has been shown to play a role in diverse cellular processes including cell proliferation, apoptosis, DNA damage response, angiogenesis, hypoxia, cell differentiation, and stem cell maintenance [28,29]. Accordingly, ING4 deficiencies have been implicated in many cancer types and several non-neoplastic diseases [30,31]. In breast cancer, *ING4* is deleted and/or downregulated in up to 34% of tumors correlating with advanced stage, lymph node positivity, and poor patient outcomes [32–34]. Expression of a dominant negative ING4 mutant increased mammary tumor metastasis in mice, corroborating the tumor suppressive function of ING4 in breast cancer [35]. ING4 has been shown to inhibit NF-kB in various cancer types including breast cancer, melanoma, glioma, squamous cell carcinoma, and hepatocellular carcinoma [30]. Dysregulated NF-kB due to ING4 deficiencies resulted in aggressive tumor behaviors such as increased cell survival, migration, and angiogenesis [33,36–39]. Since NF-kB is a mediator of immune response [40], the ING4/NF-kB axis is likely to elicit tumor-immune modulation resulting in a TIME that may contribute to disease progression, but the immune composition in ING4-deficient tumors has not been characterized to date.

In this study, we investigated the TIME in ING4-deficient breast cancer by characterizing patient samples and mouse mammary tumors. The immune markers associated with *Ing4*-deleted mouse mammary tumors were evaluated for patient survival using the METABRIC breast cancer gene expression dataset [41,42]. The study results provide the first evidence that ING4-deficient tumors harbor an immune-evasive tumor microenvironment that contributes to metastatic progression and poor patient survival in breast cancer.

## Materials and methods

### Tumor Tissue Microarray (TMA)

TMAs containing double or triple punches of 249 independent breast tumor samples were described previously [33]. In brief, TMAs were constructed by extracting 0.6 mm diameter cores from the "donor" tumor tissue blocks and transferring tissue cores into a "recipient" paraffin block. TMAs contained 598 tissue punches from 249 independent tumor samples. TMA sections of a 5mm thickness were used for immunohistochemical staining.

### Immunohistochemistry (IHC)

IHC staining of TMA sections was performed using BOND-MAX autostainer (Leica Microsystems, Wetzlar, Germany) as described previously [33,43]. Antibodies used for IHC were against ING4 (1:100, BTIM-4, Millipore Sigma), pp65RelA(S276) (1:40, Cell Signaling Technology (CST), Danvers, MA), CD4 (1:500, Abcam, Cambridge, MA), and PD-1 (1:500, Abcam). Pre-dilute antibodies for CD68 and CD8 were purchased from Leica Microsystems and used according to the manufacturer's instructions. IHC scores for ING4 and pp65/RelA staining obtained in the previous study [33] were used to correlate with the presence of the immune markers in this study. IHC staining for CD68, CD4, CD8, and PD-1 was scored manually by two trained personnel including a board-certified pathologist (J.L) in a quartile scale of 0 to +3. Each tumor was assigned an IHC score by averaging the scores of double or triple punch samples on TMA sections. IHC scores ≥1.5 were considered positive for the marker. Due to the limited tissue availability left on TMAs, the percentage of evaluable tumor IHC staining was 76–77% (189–193 out of 249 tumor samples) for CD68, CD4, and CD8, and 46% (115 out of 249 tumor samples) for PD-1.

### Mice

Female FVB/n mice were purchased from Jackson Laboratory (Bar Harbor, ME) and housed in static micro-isolation cages in a pathogen-free facility at the University of Arizona Animal Care (UAC) facility fully accredited by the Association for the Assessment and Accreditation of Laboratory Animal Care (AAALAC). Mice were provided with water and irradiated feed. All handling of mice, including mouse transfer to clean cages, was done in a Class II biosafety cabinet. Cages were checked daily, 7 days a week, by the UAC facility staff and veterinary care was provided by the UAC veterinarians as needed. All experiments including procedures, monitoring, and endpoint criteria were approved by the Institutional Animal Care and Use Committee (IACUC) at the University of Arizona (protocol #16–107). Subcutaneous injection of cells into the mammary fat pads was carried out using 21 ½ gauge needle syringes under inhalant anesthesia, 2–4% isoflurane in oxygen. Mice were placed on a heating pad during the quick recovery from anesthesia. Injectable carprofen (2-5mg/kg, Putney Inc. Portland, ME) was available to alleviate pain as needed. Mice were monitored for tumors every 2–3 days; tumors were measured at the longest axis with a caliper. Mice with tumors ≤2–2.5 cm in diameter or with one or more moribund conditions were euthanized. The range of endpoint tumor

size was approved with the consideration that caliper measurements of the longest axis over the skin fur of live animals approximated tumor diameters. Tumor growth was rapid in this study model and reached the maximum tumor size in ~3 weeks, both of which, the tumor size and duration, were necessary to allow distant metastasis to occur. Tumor mice at the endpoint criteria showed no signs of distress or discomfort, exhibiting normal behaviors of grooming, eating, drinking, and mobility. Moribund conditions for euthanasia included the following: greater than 10% weight loss, inactivity (failure to move freely, to eat or to drink), diarrhea, labored respiration, excessive sensitivity to touch or handling, rash and/or skin desquamation that interferes with normal activity and does not respond to conservative treatment. Euthanasia was carried out by carbon dioxide inhalation followed by cervical dislocation as a secondary method to ensure death, in compliance with the recommendation of the American Veterinary Medical Association.

## Generation of the p53MT mouse mammary tumor cell line

FVB.$p53^{LoxP/LoxP}$ mice, the $p53^{LoxP/LoxP}$ transgenic mouse line [44,45] bred into the FVB strain background [46] were kindly provided by Dr. Inge (Norton Thoracic Institute, St. Joseph's Hospital and Medical Center, Phoenix, AZ). Mammary glands isolated from two FVB.$p53^{LoxP/LoxP}$ homozygous female mice were processed to generate primary mammary epithelial cells (MECs) as described previously [35]. MECs were incubated overnight in the F12:DMEM MEC media containing 10% fetal bovine serum (FBS), 1x Penn-Strep antibiotics, 10 ng/ml murine EGF (Thermo Fisher Scientific, Waltham, MA), 10 ng/ml bovine pancreas insulin (Sigma-Aldrich, St. Louis, MO), and 10 mg/ml hydrocortisone (Sigma-Aldrich). Adherent cells were infected with 5 x $10^6$ plaque forming units of adeno-Cre virus (purchased from University of Iowa adenoviral core) in the F12:DMEM media containing 2% FBS for 6 hours as described previously [47]. MECs were placed in the fresh MEC media and incubated for 2 days before trypsinized and washed with Dulbecco's phosphate-buffered saline (DPBS, Thermo Fisher Scientific); 5 x $10^5$ cells in 50–100 mls DPBS ($Ca^{2+}$/$Mg^{2+}$ free, Thermo Fisher Scientific) were implanted into the #4 mammary fat pads of 21 days-old FVB/n female recipient mice (Jackson Laboratory) via subcutaneous injection using 21½ gauge needle syringes under inhalant anesthesia, 2–4% isoflurane in oxygen. A single tumor developed out of 4 implants and was harvested at 6 weeks post-implant when the tumor diameter reached ~2 cm by caliper measurement. Tumor was mechanically minced, digested with 10 mg/ml collagenase IV (Worthington Biochemical, Lakewood, NJ) in the RPMI media containing 2% FBS and 1x Penn Strep (Thermo Fisher Scientific) for 40 minutes at 37°C shaking, and washed several times until the wash was clear. Digested tumor was plated in multiple 10 cm dishes in the MEC media, incubated for 7 days, and cryo-preserved in the MEC media containing 20% FBS and 10% DMSO (Sigma-Aldrich). These cells were referred as p53MT p.1 (p53 mammary tumor passage 1). The p53MT cell line was established by passaging p53MT p.1 for additional 10 days (p53MT p.2 and p.3). p53MT p.3 cells were subsequently maintained in the F12:DMEM media supplemented with 10% FBS.

## CRISPR/CAS9-mediated deletion of the mouse *Ing4* gene

Two guide RNA (gRNA) sequences targeting the exon 1 and exon 3 of the mouse *Ing4* gene were selected according to the orthologous sequence in the human *ING4* gene [48]. The gRNA sequences were: R1, 5'-GATGGCTGCTGGGATGTATT; and M1, 5'-CTGAATATATGAGT AGCGCC. Custom-made oligonucleotides (Thermo Fisher Scientific) were cloned into the lentiCRISPRv2 plasmid (Addgene, Watertown, MA), referred as "v2" here, following the Zhang lab protocol [49,50]. Lentiviral particles containing v2, v2R1, or v2M1, were produced in

HEK293(F)T cells (American Type Culture Collection, Manassas, VA) using the pVSVg and psPAX2 viral packaging plasmids (Addgene) and Effectene transfection agent (Qiagen, Germantown, MD) as described previously [48]. p53MT cells were infected with viral particles in the presence of 2 mg/ml polybrene (Sigma-Aldrich) and were selected in the F12:DMEM media containing 10% FBS and 2 mg/ml puromycin (Sigma-Aldrich) for 10–14 days.

### p53MT tumor implants in mice

Recipient mice were 21–23 days old female FVB/n mice purchased from Jackson Laboratory. Fifty thousand p53MT v2, v2R1, or v2M1 cells in 50–100 μl sterile PBS (Ca$^{2+}$Mg$^{2+}$ free) were injected subcutaneously into the #4 mammary fat pads of recipient mice using 21 ½ gauge needle syringes under inhalant anesthesia, 2–4% isoflurane in oxygen. A total of 30 mice were implanted with p53MT cells, 10 mice per group (v2, v2R1, v2M1). Tumor size was measured at the longest axis using a caliper every 2–3 days in 21 mice that grew tumors. Four tumor mice developed skin lesions due to overgrooming and were euthanized before the endpoint was reached as part of the effort to alleviate suffering. Tumor and lung were harvested from mice at the endpoints, minced to ~1 mm$^3$ tissue bits, resuspended in a cryopreservation media containing 10% DMSO and 20% FBS in F12:DMEM, and stored in a -80˚C freezer until ready for subsequent analyses.

### Quantification of lung metastases in mice

Lung tissue bits cryopreserved at the time of tumor harvest were thawed in a 37˚C water bath, washed with the RPMI media containing 2.5% FBS and 1x Penn Strep (wash media), and minced with No. 2 scalpel for 2 minutes. Lung tissue bits were resuspended in the wash media containing 1 mg/ml collagenase I (Worthington) and 1 mg/ml collagenase IV (Worthington) and incubated at 37˚C shaking for 40–50 minutes. Cells were washed 3 times and plated in the F12:DMEM media supplemented with 10%FBS and 1x Penn Strep for 3 days. Adherent cells were trypsinized and plated in two 10 cm dishes in the media containing 2 mg/ml puromycin for 7 days. Cells were counted (1–10 x 10$^6$ cells yield per lung tissue) and 10$^3$ cells were plated in 60 mm dishes in triplicates in the media containing 2 mg/ml puromycin for 14 days. Cell foci were fixed with ice cold methanol (Fisher Scientific, Pittsburg, PA) for 10 minutes and stained with 0.5% crystal violet (Thermo Fisher Scientific) in 25% methanol at room temperature for 10 minutes. Plates were washed with dH$_2$O until the rinse was clear, dried overnight, and foci were counted.

### Western blot

Cell lysate fractionation and Western blot were performed as described previously [33]. Antibodies used in Western blots were against: ING4 (BTIM-4 hybridoma supernatant, 1:4 [33]), pp65/RelA (Ser536) (93H1, 1:1,000; CST), histone H3 (1:5000, CST), and a-tubulin (DM1A, 1:5000, Sigma-Aldrich). HRP-conjugated anti-mouse and anti-rabbit secondary antibodies were used (1:5000, Thermo Fisher Scientific) and detected using Immobilon ECL Ultra Western HRP Substrate reagents (Millipore Sigma, St. Louis, MO). Western blot images were acquired and analyzed using the LI-COR Odyssey Fc Imaging System and the LI-COR software Image Studio Lite (LI-COR Biosciences, Lincoln, NE).

### Flow cytometry (FACS)

Fluorescence-conjugated antibodies used in FACS were purchased from BioLegend (San Diego, CA). Antibodies in the myeloid and NK cell panel were: PerCP-Cy5.5/CD45 (30-F11 clone, 1:1000), FITC/CD11b (M1/70 clone, 1:1000), PE/CD274 (10F.9G2, 1:40), BV421/Granzyme b

(QA18A28 clone, 1:20), PE-Cy7/NKp46 (29A1.4 clone, 1:20), APC/Gr-1 (RB6-8C5 clone, 1:800), and BV510/F4/80 (BM8 clone, 1:40). Antibodies in the T cell panel were: PerCP-Cy5.5/CD45 (30-F11 clone, 1:1000), PE-Cy7/CD3ε (145-2C11 clone, 1:40), APC/FoxP3 (Thermo Fisher Scientific, FJK-16s clone, 1:20), BV421/Granzyme B (QA18A28 clone, 1:20), PE/PD-1 (29F.1A12 clone, 1:20), FITC/CD8α (53–6.7 clone, 1:50), and BV510/CD4 (GK1.5 clone, 1:40). In addition, Fixable Viability Stain (BD Biosciences, Bedford, MA, 1:2000) was used for dead cell exclusion and FcγR III/II receptors were blocked with FcX TruStain (BioLegend) using the manufacturer's instructions. To obtain a single cell suspension for flow cytometry analysis, minced tumors were digested with 2 mg/ml collagenase IV (Worthington) and 20 IU/ml Roche DNase I (Millipore Sigma) by incubating for 20–30 minutes in a 37°C shaking water bath. To enable intracellular staining, True-Nuclear Transcription Buffer Set (BioLegend) was used to prepare cells according to the manufacturer's specifications. UltraComp eBeads (Thermo Fisher Scientific) were used for compensation. Unstained and Fluorescence-Minus-One controls were used as needed. FACS samples were run on Canto II (BD Biosciences) and analyzed using FlowJo (BD Biosciences, v10.7.1) software. Samples were gated on FSC vs. SSC to remove debris, single cells were selected by gating on FSC-A vs. FSC-H, and dead cells were excluded using live/dead staining.

## Reverse transcription quantitative PCR (RTqPCR)

RTqPCR was performed using Taqman Gene Expression Assays (Thermo Fisher Scientific) with FAM-labeled probes for *mIL6* (Mm00446190_m1) and VIC-labeled *mGAPDH* (Mm99999915_g1). Fluorescence probe amplification was detected by QuantStudio™ 6 Flex Real-Time PCR System (Thermo Fisher Scientific). Data were analyzed by DDCt calculation normalized to *mGAPDH*.

## Transwell migration assay

Cell migration assays were performed using transwell inserts as described previously [48]. In brief, 50,000 cells were placed in the inserts containing a semipermeable membrane with 8 μm-sized pores (Fisher Scientific). Recombinant TNFa (R&D Systems, Minneapolis, MN) or IL-1b (R&D Systems) was dissolved in PBS to a 10 mg/ml 1000x stock concentration. Migrated cells on the bottom side of the membrane were fixed using 100% cold methanol (Fisher Scientific), stained with 4′,6-diamidino-2-phenylindole (DAPI, Vector Labs, Burlingame, CA), and visualized under a fluorescence microscope. Cell numbers were determined by averaging cell counts from a minimum of 6 field images per membrane.

## METABRIC gene expression dataset and statistical analysis

The METABRIC [41,42] dataset was downloaded from cBioPortal for Cancer Genomics (www.cbioportal.org). The mean expression values of each gene were used as the cut-offs for comparing patient survival. Kaplan-Meier survival analyses were performed using the Graph-Pad Prism 8 software and the log-rank test (GraphPad Software, San Diego, CA); $p < 0.05$ was considered significant. For other analyses, Fisher's Exact Probability test, Student t-test, or One-way ANOVA was used to determine statistical significance; $p < 0.05$ was considered significant or specific $p$ values were shown in the figures when appropriate.

## Results

### Immune cell markers in ING4-deficient breast tumors

Studies have shown that ING4 inhibits NF-kB in various cancer types [30,31]. In breast cancer, NF-kB activation was prevalent among ING4-deficient tumors, associated with advanced

stage, lymph node positivity, and poor patient survival [33]. Since NF-kB is a key transcription factor that mediates inflammatory response including cytokine production [40], we hypothesized that ING4-deficient tumors may harbor an immune microenvironment resulting from aberrant NF-kB activity, which in turn contributes to aggressive tumor progression. To determine whether specific immune cell types were enriched in the ING4-deficient tumor microenvironment (TME), we first employed immunohistochemical (IHC) staining of breast tumor tissue microarrays (TMAs) for selective immune markers: CD68 (macrophages), CD4 (helper T cells), CD8 (cytotoxic T cells) and PD-1 (an immune checkpoint molecule on lymphocytes). These markers were chosen because of the well-documented roles for macrophages and exhausted T cells in breast cancer pathogenesis [51,52].

The TMAs used in this study have been characterized previously, consisting of 246 independent primary breast tumors [33]. We stained the TMA subset (n = 198) because of the limited tissue availability, with the antibodies against CD68, CD4, CD8, and PD-1, and correlated each immune marker with ING4 and/or pp65/RelA(Ser276) (an activated marker for the p65 subunit in the p65/p50 NF-kB heterodimer) staining. Examples of the positive IHC staining for each antibody are shown in Fig 1A (a, b, and c are the sections from different tumors; c-f are the serial sections from the same tumor). IHC staining was scored in a scale of 0–3 and each tumor was assigned a number by averaging IHC scores between double or triple punches on TMAs as described previously [33]. An average score ≥1.5 was considered positive for the marker; the percentages of tumors positive for each marker are shown in Fig 1B. The results showed that 37% of tumors (92/246) were ING4-deficient and 34% (84/245) were positive for pp65/RelA. Notably, 89% of tumors (171/193) were positive for CD68, indicating the ubiquitous presence of macrophages in breast tumors. In contrast, T cells were present in tumor subsets, 26% for CD4 (50/193) and 40% for CD8 (75/189). Tumors containing PD-1$^+$ leukocytes consisted of 18% (24/132).

We also determined the percentage of tumors that were double- or triple-positive for the immune markers. The results showed that tumors positive for all three CD68, CD4, and CD8 markers comprised of 20.4% (38/186), while tumors negative for all three markers were 8.6% (16/186, Fig 1C). Tumors positive for CD68$^+$ macrophages without T cell presence consisted of 47.8% (89/186). In contrast, tumors positive for CD4$^+$ or CD8$^+$ T cells without CD68+ macrophages were rare, comprising less than 1% (Fig 1C). The Venn diagram analysis showed that CD68$^+$ macrophages and CD8$^+$ T cells were present in the same tumors irrespective of CD4$^+$ T cells, whereas CD4$^+$ T cells were present when both CD68$^+$ and CD8$^+$ cells were present (Fig 1C). Whether PD-1$^+$ leukocytes were prevalent in tumors positive for CD4$^+$ or CD8$^+$ T cells could not be determined because the number of PD-1$^+$ tumors was small.

We examined if any of the four immune markers was associated with ING4 or pp65/RelA. As shown previously [33], a significantly high proportion of ING4-low tumors were positive for pp65/RelA (44% vs 28%, $p$ = 0.018, Fig 1D left graph), indicating that ING4-deficient tumors have a propensity for NF-kB activation [33]. In comparison, CD68, CD4, CD8, or PD-1 was not associated with ING4 or pp65/RelA (Fig 1D). These results indicated that ING4 deficiency or NF-kB activation alone may not result in a TME enriched for macrophages or specific T cell types.

We next evaluated whether the combination of ING4 deficiency and NF-kB activation together fostered recruitment of macrophages or T cells. The results showed a significant association between ING4-low/pp65-high tumors and CD8$^+$ T cells (58.6% vs 31.6%, p = 0.046, Fig 1E). The PD-1 positivity was lower in this ING4-low/pp65-high tumor cohort but statistical significance could not be determined due to the small cohort size (Fig 1E). In addition, double staining for CD8 and PD-1 could not be performed due to the limited availability of tissue

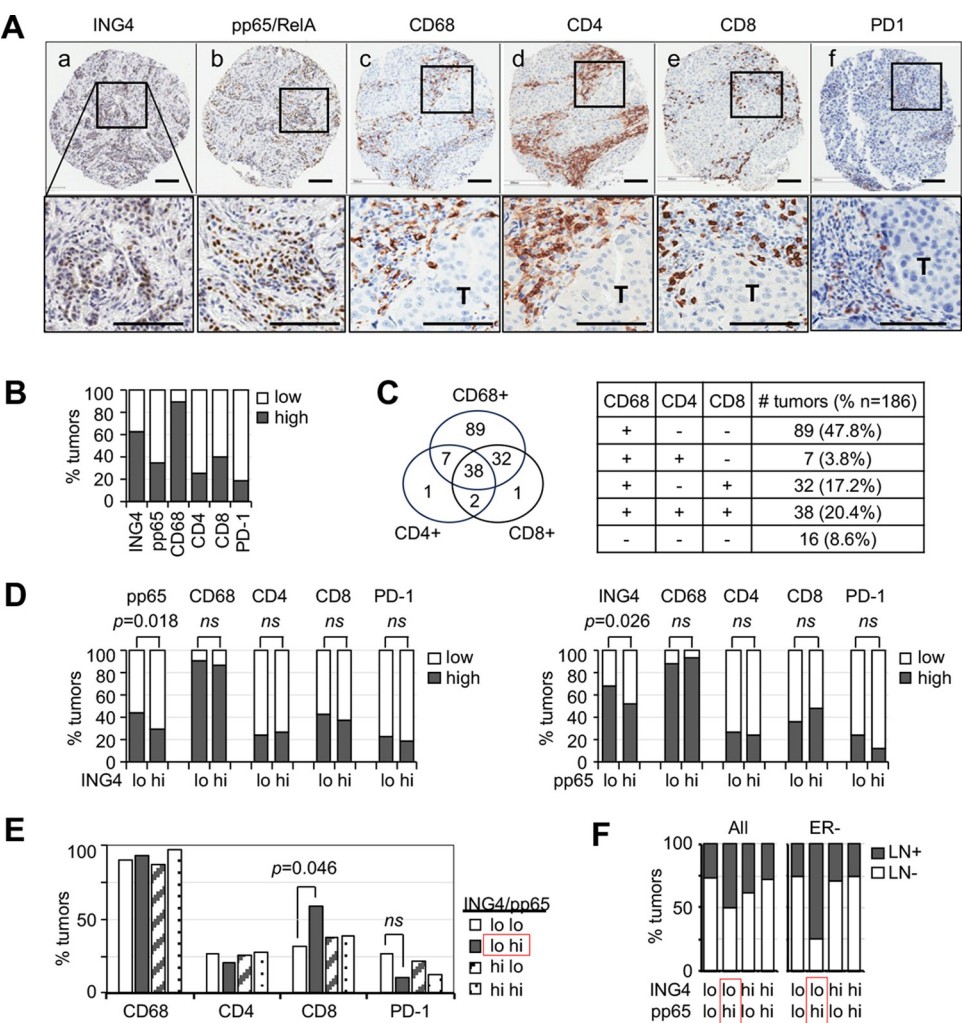

**Fig 1. ING4, NF-kB, and immune markers in breast tumors. (A)** Immunohistochemical staining images of tumors positive for: a) ING4 (IHC score +1), b) pp65/RelA(Ser276) (IHC score +2), c) CD68 (IHC score +2), d) CD4 (IHC score +3), e) CD8 (IHC score +2), f) PD1 (IHC score +3); black boxes in the top panel are presented in higher magnification images in the bottom panel; black scale bars denote 100 mm in length. **(B)** Percentage of tumors positive for each marker (solid dark bar): ING4 (154/246, 63%), pp65 (84/245, 34%), CD68 (171/193, 89%), CD4 (50/193, 26%), CD8 (75/189, 40%), and PD-1 (24/132, 18%). **(C)** Venn diagram and table showing the number of tumors positive for CD68, CD8, CD4, and/or PD-1. **(D)** No association between immune markers with ING4-low vs ING4-high (left graph) or NF-kB-low vs NF-kB-high (right graph) tumors. **(E)** Increased CD8+ tumors in ING4-low/pp65-high tumors (solid dark bar, 59% vs 32–38%); p values were determined by Fisher's Exact test; *ns*, not significant. **(F)** High lymph node positivity in ING4-low/pp65-high tumors (LN+, solid dark bar): All, all tumors; ER-, estrogen receptor-negative.

sections. Thus, it is presently not known whether CD8$^+$ T cells in ING4-low/pp65-high tumors exhibit PD-1$^+$ immune exhausted phenotype.

We next examined the clinical significance of ING4-low/pp65-high status by evaluating an association with the lymph node positivity. The results showed that a high percentage of ING4-low/pp65-high tumor patients were lymph node-positive, compared to the other ING4/pp65 expression combination tumor groups (50% vs 26–38%, Fig 1F). This association between the ING4-low/NF-kB-high status and lymph node positivity was more pronounced in estrogen receptor-negative (ER-) breast cancer (75% vs 25–29%, Fig 1F, right panel). In summary, the breast tumor IHC results showed that ING4-low/pp65-high tumors were associated with CD8$^+$ T

cells and lymph node metastasis in ER- breast cancer. The presence of CD8[+] T cells correlating with metastasis is counterintuitive, as CD8[+] T cells are generally associated with a tumoricidal activity and good prognosis [17,18]. It may indicate that the CD8[+] T cells in ING4-low/NF-kB-high tumors are dysfunctional and/or tumor promoting but requires characterization.

## Aberrant NF-kB activity in *Ing4*-deleted mouse mammary tumor cells

To better characterize the TIME associated with metastatic ING4-deficient breast cancer, we sought to utilize an immune-competent mouse model of ER- breast cancer. Mouse models of triple-negative breast cancer (TNBC) exist, namely 4T1 and 66.1, that are widely used. Both cell lines were derived from a single mammary tumor that arose spontaneously in a BALB/cfC3H strain mouse that carried mouse mammary tumor virus, MMTV [53]. Tumors resulting from these cell implants were fast-growing, spontaneously metastasized to distant sites, and were associated with immune-suppressive phenotypes [54–57]. Moreover, the recent multi-omics study of 4T1 tumors showed numerous driver gene mutations as well as complex epigenetic changes [58]. Because of their complex genetic background and spontaneous metastasis, we reasoned that deciphering a single gene effect on the immune TME associated with metastasis may be difficult in these models without a large mouse cohort size and systematic multivariate analyses. Thus, we decided to devise a TNBC mouse model with a simpler genetic background utilizing the *p53^{loxP/loxP}* conditional knock-out transgenic mouse line [44,45]. The choice of *p53* deletion for modeling TNBC was because *TP53* tumor suppressor mutations are the single most prevalent driver mutation in the TNBC subtype of breast cancer [59].

To establish a *Tp53*-deleted mammary tumor model, we isolated primary mammary epithelial cells (MECs) from two FVB.*p53^{loxP/loxP}* homozygous female mice and transduced the Cre recombinase using adeno-Cre virus in culture (see Method). One million MECs were implanted into the mammary fat pads in four recipient FVB mice, from which a single mammary tumor developed in 6 weeks. The tumor was dissociated and cultured for 17 days, establishing a cell line, herein referred to as p53MT (p53-deleted mammary tumor). Subsequent orthotopic implants of $10^5$ p53MT cells grew to tumors of >2cm diameter in 3 weeks but no lung metastatic nodules were detected under a dissecting microscope examination, providing an appropriate model to investigate the metastatic and immune TME effects of ING4 deficiencies in TNBC.

The mouse *Ing4* gene in p53MT cells was deleted using a CRISPR/CAS9 system (see Method). Guide RNA sequences targeting the *Ing4* gene at exon 1 and exon 3 (R1 or M1, a schematic diagram in Fig 2A) were cloned into the lentiCRISPRv2 plasmid and used to establish the cell lines, namely p53MT v2 (the vector control), v2R1, and v2M1. Western Blot showed >99% and >90% reduction of the Ing4 protein in v2R1 and v2M1 cells, respectively (Fig 2B). To determine whether *Ing4*-deleted cells showed aberrant activation of NF-kB, v2, v2R1, or v2M1 cells were treated with TNF-a or IL-1b for 1 hour in serum free media. Western blot showed increased pp65/RelA(Ser536) in the nuclear fraction of *Ing4*-deleted v2R1 or v2M1 cells treated with TNFa or IL-1b compared to v2 cells, indicating increased activation of NF-kB in *Ing4*-deleted cells (Fig 2B). Additionally, *Il6*, an NF-kB transcription target gene, was aberrantly expressed in v2R1 or v2M1 cells compared to v2 cells in response to TNFa and/or IL-1b (Fig 2C). Lastly, TNFa or IL-1b induced migration of *Ing4*-deleted v2R1 and v2M1 cells, but not of the vector control v2 cells (Fig 2D). These results demonstrated that *Ing4* deletion in p53MT cells resulted in aberrant activation of NF-kB.

## Increased penetrance and metastasis of *Ing4*-deleted mouse mammary tumors

To determine whether increased migration of *Ing4*-deleted cells *in vitro* correlated with tumor metastasis *in vivo*, we implanted v2, v2R1, or v2M1 p53MT cells into the mammary fat pads of

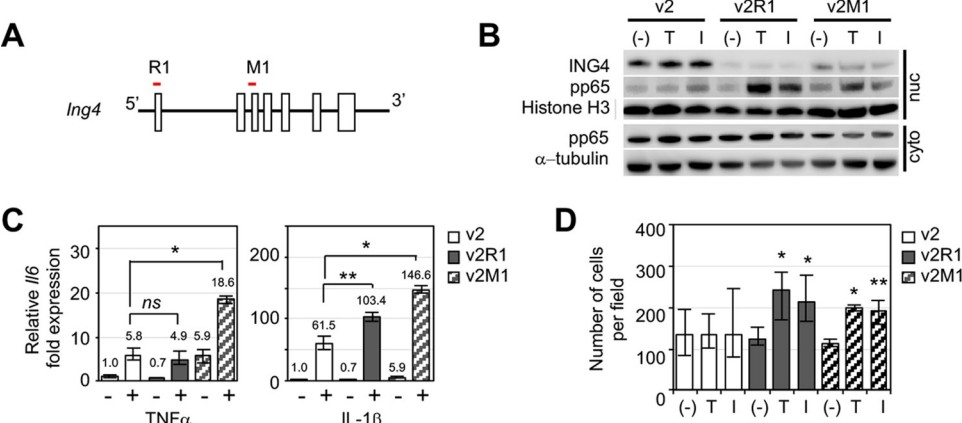

**Fig 2. Aberrant activation of NF-kB in *Ing4*-deleted p53MT mouse mammary tumor cells. (A)** Schematic diagram of the *Ing4* genome with exons (open boxes); CRISPR/CAS9 gRNAs R1 and M1 mapped to exon 1 and exon 3 (red line). **(B)** Nuclear accumulation of pp65/RelA in *Ing4*-deleted p53MT cells treated with cytokines: v2, the vector control cells; v2R1, R1 gRNA construct cells; v2M1, M1 gRNA construct cells; cells were treated with PBS (-), 10ng/ml TNFa (T), or 10ng/ml IL-1b (I), for 1 hour in serum-free media prior to cell fractionation; nuc, nuclear fraction; cyto, cytoplasmic fraction; Western blot for ING4 and phospho-p65/RelA (ser536); Histone H3 and a-tubulin antibodies were used as the loading controls. **(C)** Relative fold expression of the mouse *Il6* gene in p53MT cells treated with (+) or without (-) TNFa (left panel) or IL-1b (right panel) for 4 hours in serum-free media. Numbers denote the average fold change of a minimum three replicates: open bar, v2; closed bar, v2R1; serrated bar, v2M1; RT-qPCR to quantify *Il6* expression using *Gapdh* as the reference; *p* values were determined by two-tailed student t-test; *ns*, not significant. **(D)** Cytokine-induced migration of *Ing4*-deleted p53MT cells: PBS (-),TNFa (T), or IL-1b (I); v2, v2R1, or v2M1 cells migrated through 8mm pores in the transwell inserts were stained with DAPI and visualized under a fluorescent microscope. Cell numbers were determined by averaging a minimum of 4 images per experiment from at least 3 independent experiments; *p* values were determined by two-tailed student t-test; *, $p < 0.001$; **, $p < 0.05$.

recipient mice. In a cohort of 10 mice each group, 50% of v2 implanted mice developed tumors, whereas 80% of v2R1 or v2M1 implanted mice did by 36 days of post-implant follow up (Fig 3A). Comparable growth rates were observed by comparing the longest axis measurements of v2, v2R1 and v2M1 tumors using a caliper in the mice that formed tumors (n = 4 each group shown in Fig 3B). Resected tumors at the endpoints were within the IACUC approved size range. These results indicated that *Ing4* deletion did not impact tumor growth but increased tumor penetrance. Increased tumor penetrance suggested that *Ing4*-deleted cells may be better "fit" to form tumors potentially via cell survival mechanisms including immune evasion (see below and Discussion).

We compared tumor metastasis to the lung between v2, v2R1, and v2M1 tumor mice. As tumor metastases in the lungs were not readily visible under a dissecting microscope, we devised a protocol to quantify tumor cells disseminated to the lung tissues by modifying Pulaski et al. [55]. In brief, the lungs harvested from tumor-bearing mice were mechanically dissociated, digested with collagenases I and IV, and plated in the media containing puromycin for selection of the CRISPR constructs in tumor cells. After 3 weeks, the foci were stained with crystal violet and counted. The results showed that the lungs from a wildtype mouse yielded no puromycin-resistant foci as expected, whereas 1 to 22 puromycin-resistant foci formed per plate from v2 tumor mouse lung tissues (Fig 3C). In comparison, the lungs from v2R1 or v2M1 tumor mice yielded an increased number of foci ranging from 8 to 208 in the selective media (Fig 3C), indicating an 8- to 9-fold increase. Although the statistical significance could not be reached due to the small cohort size, the trend showed that *Ing4*-deleted tumors were more metastatic, consistent with the hypothesis that ING4 deficiencies promote metastatic progression in breast cancer.

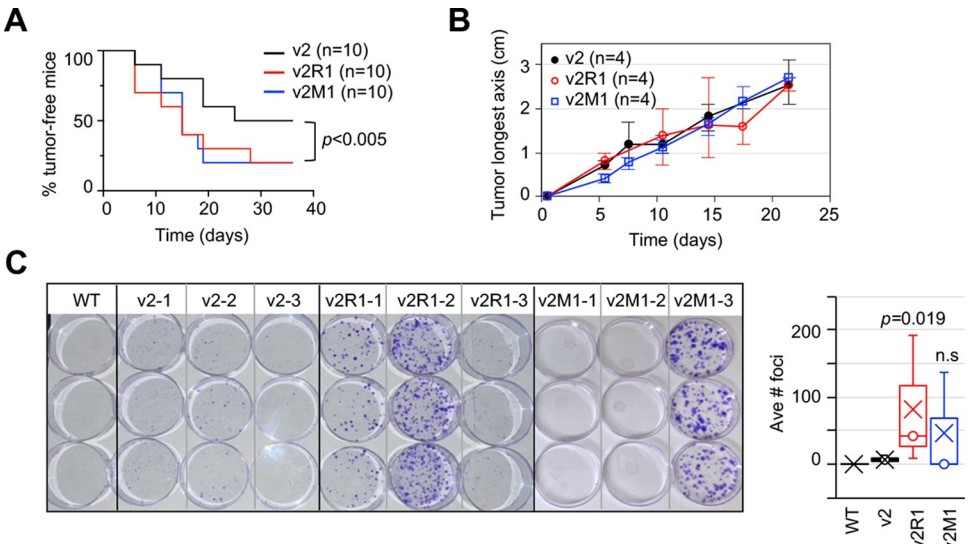

**Fig 3. Growth and metastasis of *Ing4*-deleted p53MT tumors in mice. (A)** Increased penetrance of *Ing4*-deleted tumors: Kaplan-Meier analysis of tumor-free mice during a 36 day follow up after $5 \times 10^4$ cell implant into the #4 mammary fat pads. Black line, the v2 vector control (n = 10); red line, v2R1 (n = 10); blue line, v2M1 (n = 10); *p* values determined using two-tailed unpaired student t-test, v2 vs v2R1 *p* = 0.0007, v2 vs v2M1 *p* = 0.0043 and one-way ANOVA *p* = 0.0031 **(B)** Tumor growths in representative 4 mice in each group were graphed using an average tumor measurement of the longest axis at each time points, setting day 0 for each tumor when a 1–2 mm nodule is detected. Black circle, v2 (n = 4); red circle, v2R1 (n = 4); blue square, v2M1 (n = 4). **(C)** Tumor metastasis to the lung quantified by lung tissue harvest and dissociation at end point followed by plating in the media selecting for puromycin resistance encoded by the plasmid constructs. The results from WT (n = 1, no tumor implant), v2 (n = 3), v2R1 (n = 3), and v2M1 (n = 3) tumor mice are shown: puromycin-resistant foci (crystal blue-staining) were counted and presented in a Box plot with X marks the mean value and o median. *p* values were determined by student t-test comparing to v2.

## Immune composition in *Ing4*-deleted mammary tumors

To characterize the immune components in *Ing4*-deleted mammary tumors, we employed flow cytometry and evaluated myeloid and lymphoid cell populations within the tumors. Single cells isolated from tumors were labeled with antibodies and a viability dye as described previously [60]. Live cells were gated to evaluate myeloid and natural killer (NK) cells as presented in the schematic diagram (Fig 4A): myeloid derived suppressor cells (MDSC) were defined as CD45+CD11b+Gr-1hi, whereas tumor associated macrophages (TAM) were CD45+CD11b+Gr-1loF4/80+. CD45+ cells were also gated for NKp46 and granzyme B (GzmB) to assess the NK cell population (CD45+NKp46+GzmB+) [11,52]. The results showed that the percentages of MDSC and NK cells were comparable between the v2 vector control and *Ing4*-deleted v2R1 or v2M1 tumors. In contrast, the frequency of TAMs was significantly increased by 2- or 3-fold in *Ing4*-deleted tumors (Fig 4B). These results suggested that TAMs may contribute to ING4-deficient tumor progression but will require further characterization (see Discussion).

PD-L1 expression in tumors or in TAMs has been shown as a mechanism of the immune-suppressive TME [51,61]. Thus, we determined the percentage of PD-L1+ cells in the tumor/stroma compartment (CD45- cells) as well as in the intratumoral immune compartment (CD45+ cells). The results showed no differences in PD-L1+ cell populations between *Ing4*-deleted and the vector control tumors (Fig 4C). These data suggested that PD-L1-mediated immune suppression may not be the major mechanism contributing to the aggressiveness of *Ing4*-deleted tumors.

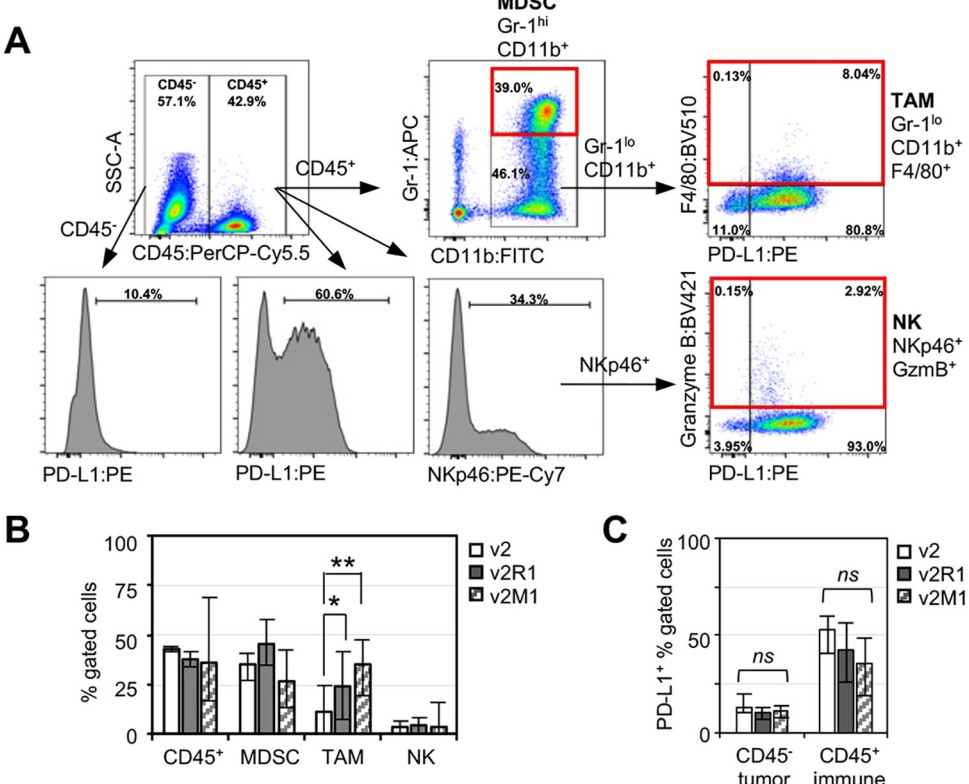

**Fig 4. Myeloid and natural killer cells in *Ing4*-deleted p53MT tumors. (A)** Gating strategy for myeloid and natural killer cells. Numbers indicate the percentage in the gate. v2-3T, a v2 tumor; MDSC, myeloid derived suppressive cells; TAM, tumor associated macrophages; NK, natural killer cells. **(B)** Average percentage of the marker positive cells in v2 (n = 4), v2R1 (n = 3), v2M1 (n = 5) tumors. Bar graphs display the percentage of CD45$^+$ cells out of total live cells; percentage of Gr-1$^{hi}$CD11b$^+$ (MDSC) out of total CD45$^+$ cells; percentage of F4/80$^+$ (TAM) out of CD45$^+$Gr-1$^{lo}$CD11b$^+$ cells; and percentage of GZMB$^+$ (NK) cells out of total CD45$^+$NKp46$^+$ cells. Open bar, v2; solid bar, v2R1; serrated bar, v2M1; error bars show minimum and maximum values; *, $p<0.05$; **, $p<0.005$. **(C)** Average percentage of PD-L1 positive cells in the CD45$^-$ (predominantly tumor) or CD45$^+$ (immune) compartments. *ns*, not significant.

We next evaluated tumor samples with a T cell marker panel (Fig 5A). In brief, CD45$^+$CD3$^+$ T cells were gated for the expression of CD4 vs CD8 (Tc, cytotoxic T cells). CD4$^+$ T cells expressing FoxP3 were considered regulatory T cells (Treg) and cells lacking FoxP3 were considered helper T cells (Th) [52]. All T cell subsets were stained for GzmB and PD-1. The results showed that the percentages of CD4$^+$ or CD8$^+$ T cells were comparable between tumors (Fig 5B). Strikingly, *Ing4*-deleted tumors contained a significantly reduced percentage of CD4$^+$GzmB$^+$ T cells compared to the vector control tumors, but the percentage of CD8$^+$GzmB$^+$ T cells were comparable between tumors (Fig 5B). GzmB$^+$CD4$^+$ T cells have recently been shown to play a role in immune-mediated killing and therapy response in cancer [62–64]. Thus, these results suggested that the *Ing4*-deleted TME may be less tumoricidal with a reduced frequency of GzmB$^+$CD4$^+$ T cells, more permissive toward tumor development and progression.

It is noteworthy that the proportion of GzmB$^+$ CD4$^+$ T cells were decreased in *Ing4*-deleted tumors in both FoxP3$^-$ effector and FoxP3$^+$ regulatory CD4$^+$ T cell populations (Fig 5B). Studies have shown that GzmB$^+$FoxP3$^+$ Treg cells have activities that kill CD4$^+$ T effector cells as part of the mechanism of Treg-mediated immune suppression [65]. In this scenario, a decrease in GzmB$^+$FoxP3$^+$ Treg cells would result in an increase in Th effector cells contributing to a

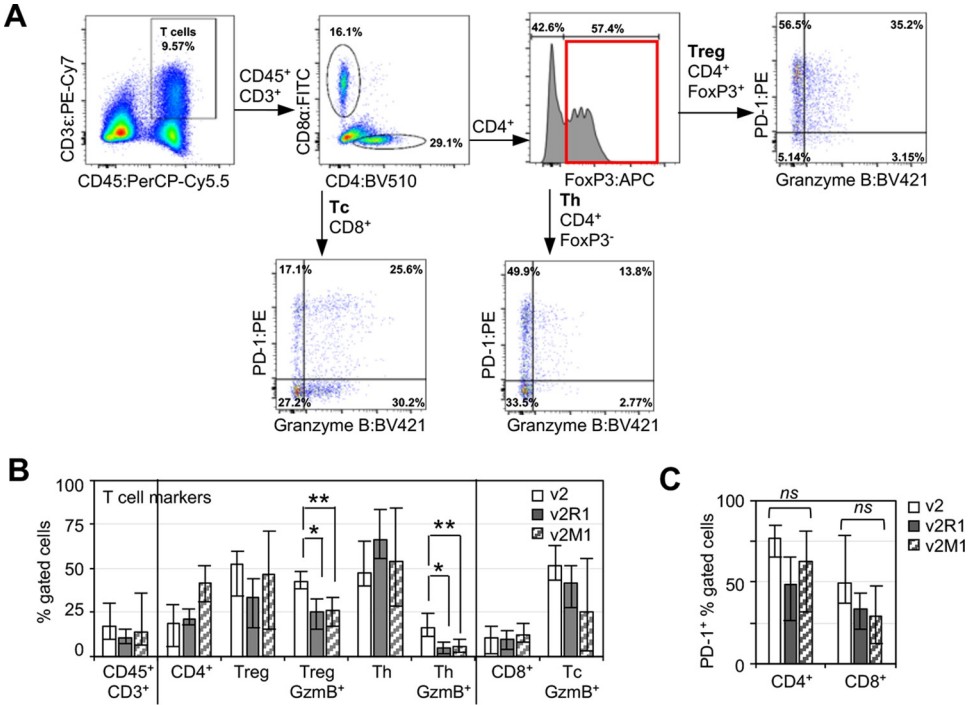

**Fig 5. T cells in *Ing4*-deleted p53MT tumors. (A)** T cell markers and the gating scheme. v2-3T, a v2 tumor; Treg, CD4$^+$ regulatory T cells; Tc, CD8$^+$ cytotoxic T cells; Th, CD4$^+$ helper T cells. Numbers indicate the percentage in the gate. **(B)** Average percentage of the marker positive cells in v2 (n = 4), v2R1 (n = 3), v2M1 (n = 5) tumors. Bar graphs display the percentage of CD45$^+$CD3$^+$ T cells out of total live cells; CD4$^+$ and CD8$^+$ T cells out of all T cells; percentage of CD4$^+$FoxP3$^+$ (Treg) and CD4$^+$FoxP3$^-$ (Th) cells out of total CD4$^+$ T cells; and percentage or GzmB$^+$ cells out of total Treg, Th, or Tc populations, as indicated. Open bar, v2; solid bar, v2R1; serrated bar, v2M1; error bars show minimum and maximum values; *, $p<0.05$; **, $p<0.01$. **(C)** Average percentage of PD-1$^+$ cells in the CD4$^+$ or CD8$^+$ T cell compartment. *ns*, not significant.

tumoricidal TME. To evaluate the opposing tumor immune effects of GzmB$^+$ Treg vs GzmB$^+$ Th cells, we determined the ratio of GzmB$^+$FoxP3$^+$/GzmB$^+$FoxP$^-$ CD4 T cells. The results showed that v2 tumors had the ratio of 2.6±0.8, whereas v2R1 and V2M1 tumors had the ratio of 5.4±2.7 and 4.8±2.2, respectively. These data indicated that *Ing4*-deleted tumors contained excess GzmB$^+$ Tregs that may contribute to the decrease in Th effector cells, resulting in an immune-evasive TME and tumor promotion.

Of note, we observed that a significant portion of T cells in tumors expressed PD-1 but showed no difference between *Ing4*-deleted and the vector control tumors (Fig 5C), indicating that *Ing4* deletion in tumors did not affect PD-1-mediated checkpoint regulation of T cells in the TME.

## Clinical significance of decreased granzyme B expression in *ING4*-deficient breast cancer

As the immune marker evaluation of *Ing4*-deleted mouse tumors showed increased TAMs (Gr-1$^{lo}$CD11b$^+$F4/80$^+$) and decreased GzmB$^+$CD4$^+$ T cells, we sought to determine the clinical significance of these findings by analyzing the METABRIC (Molecular Taxonomy of Breast Cancer International Consortium) gene expression data set [41,42]. We focused the analyses on the basal subtype tumors (n = 209) as the p53MT mouse model was derived from *Tp53* deletion, the molecular lesion associated with TNBC [59]. We stratified patient samples using

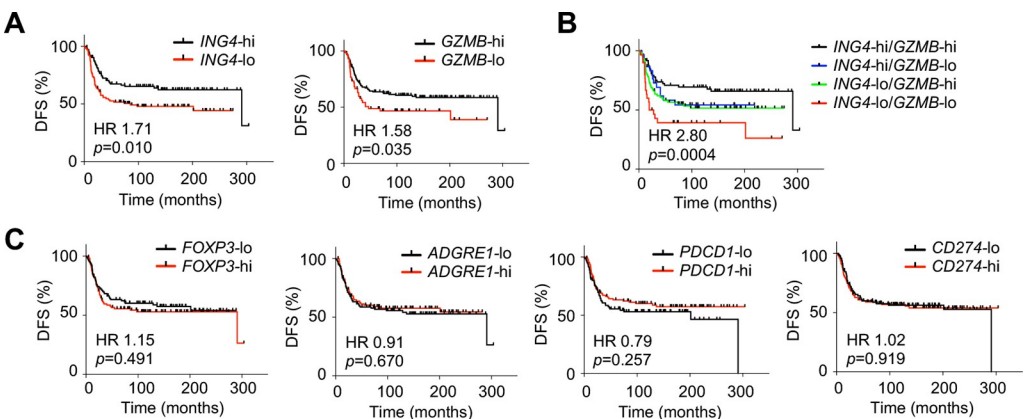

**Fig 6.** *ING4* and *GZMB* **expression associated with patient survival in breast cancer.** Kaplan-Meier survival analyses of the METABRIC data set using the mean gene expression values as the cut-off in the basal subtype breast cancer (n = 209). **(A)** *ING4*, *GZMB* (the granzyme B gene). **(B)** combination of *ING4* and *GZMB* expression levels; black line, *ING4*-high/*GZMB*-high (n = 74); blue line, *ING4*-high/*GZMB*-low (n = 30); green line, *ING4*-low/*GZMB*-high (n = 74); red line, *ING4*-low/*GZMB*-low (n = 31). **(C)** *FOXP3*, *ADGRE1* (the F4/80 gene), *PDCD1* (the PD-1 gene), and *CD274* (the PD-L1 gene). DFS, disease-free survival; HR, Hazard Ratio; *p* values determined by the log-rank (Mantel-Cox) test.

the mean values of each gene expression and performed Kaplan-Meier survival analyses. The results showed that low expression of *GZMB* (the gene encoding granzyme B) was significantly associated with poor patient survival (HR = 1.58, 95% CI 0.99–2.53, p = 0.035, Fig 6A) as was *ING4*-low expression (HR = 1.71, 95% CI 1.14–2.58, p = 0.010, Fig 6A). These supported the tumor suppressive roles for granzyme B and ING4 in the basal subtype breast cancer. Strikingly, patients with tumors expressing the *ING4*-low/*GZMB*-low combination had the worst survival outcomes compared to the other gene expression combinations (*ING4*-low/*GZMB*-low vs *ING4*-high/*GZMB*-high: HR = 2.80, 95% CI 1.36–5.75, p = 0.0004, Fig 6B). These results suggested that reduced granzyme B expression exacerbated the aggressiveness of ING4-deficient breast cancer.

Expression of the other immune markers such as *FOXP3*, *ADGRE1* (the gene encoding F4/80), *PDCD1* (PD-1 gene), *CD274* (PD-L1 gene) alone or in combination with *ING4* did not correlate with patient survival (Fig 6C), highlighting the unique relationship between *GZMB* and *ING4* expression in the basal subtype of breast cancer. *ITGAM* encoding CD11b related to TAMs could not be evaluated for patient survival because the gene expression was not available in the METABRIC dataset. Further characterization of TAMs in *Ing4*-deleted mammary tumors will be necessary to ascertain the phenotypes and functional contribution of TAMs in ING4-deficient breast cancer progression.

Taken together, the METABRIC data analyses provided clinical evidence for the functional relationship between *ING4* and *GZMB* in the basal subtype of breast cancer and supported the conclusion that the decrease in GzmB$^+$CD4$^+$ T cells in the TME contributes to metastatic tumor progression of ING4-deficient breast cancer.

## Discussion

The results presented in this study provided the first evidence that ING4-deficient tumors harbor a TIME that contributes to metastatic progression and poor patient survival outcomes in breast cancer. Intriguingly, the TIME of *Ing4*-deleted mouse mammary tumors consisted of increased tumor-associated macrophages (TAMs) and decreased GzmB$^+$CD4$^+$ T cells. Low expression of the Granzyme B gene (*GZMB*) in combination with *ING4*-deficiencies was

significantly associated with poor patient survival in the basal subtype of breast cancer, supportive of the clinical significance of granzyme-B-positive immune cell depletion in ING4-deficient tumors.

Contribution of TAMs in tumor development and progression has been well documented [66]. Especially M2-polarized TAMs were shown to secrete pro-tumor factors that can directly promote tumor growth, angiogenesis, and metastasis in cancer [67]. Additionally, TAMs have been shown to orchestrate the TIME by commencing various activities such as recruiting tumor infiltrating lymphocytes (TILs), modulating the CD8+/CD4+ T cell ratio, increasing the immune suppressive Treg cell population, and upregulating immune checkpoint molecules such as PD-L1 and PD-1 [66,68,69]. Interestingly, these TAM-associated TIME phenotypes were not apparent in *Ing4*-deleted tumors, indicating that the phenotypes and functional consequences of TAMs in *Ing4*-deleted tumors are yet to be defined.

*Ing4*-deleted mouse mammary tumors contained a significantly reduced number of GzmB+CD4+ T cells. Granzyme B (GzmB) is a member of the serine protease family, that immune cells directly inject into the target cells via perforin-formed pores following MHC-mediated cell-cell conjugation [70]. GzmB in the target cell cytoplasm cleaves and activates caspases inducing apoptosis, thus providing a powerful arm of cytocidal immune response during infection and disease [70,71]. Granzymes have been considered the main arsenal of CD8 cytotoxic T lymphocytes (CTLs) and Natural Killer (NK) cells [71]. In comparison, GzmB+CD4+ T cells, also known as CD4 CTLs, were mainly associated with viremic states and autoimmune diseases [72]. Subsequent studies showed that CD4 CTLs could eliminate tumors in an MHC class II-dependent manner in experimental models [72,73]. More recently, scRNAseq of TILs in human bladder cancer samples revealed the presence of CD4 CTLs expressing granzymes and perforin [74]. Moreover, the CD4 CTL gene signature was correlated with anti-PD-L1 therapy response in patients, supportive of the CD4 CTL function in anti-tumor immune response [74]. Our results showing decreased CD4 CTLs in *Ing4*-deleted tumors correlating with metastasis and poor patient survival are consistent with the roles of CD4 CTLs in anti-cancer immunity.

How CD4 CTLs are regulated in the TME is not well understood. *In vitro*, CD4 CTLs could be induced from naïve T cells as well as from other CD4+ T subtypes (e.g. Th1, Th2, Th17, Treg) by several cytokines and MHC class II-restricted antigens, indicating multiple routes of CD4 CTL maturation [75–77]. Our study presents a novel mechanism of CD4 CTL fate regulation driven by ING4 in a cell non-autonomous manner. How tumor expression of ING4 regulates the CD4 CTL fate in the TME is unclear. Potentially, ING4 could regulate the expression of cytokines required for CD4 CTL lineage differentiation; the absence of ING4 renders CD4 CTL differentiation less effective. Investigation of ING4-regulated cytokines may provide insight into the GzmB+CD4+ T cell differentiation mechanism(s).

Another possible mechanism of ING4 may involve the regulation of MHC class II expression in tumors. Tumor expression of MHC class II has been associated with favorable patient outcomes in several cancers including breast cancer [78,79]. In the absence of ING4, downregulation of MHC class II may occur in tumors, resulting in limited CD4 CTL activation in the TME. Of note, elevated MHC class II expression in lymph node metastases was correlated with increased Tregs resulting in decreased Th cells [80]. Since *Ing4*-deleted tumors in our study did not show an increase in Tregs, MHC class II-mediated Treg expansion does not appear to be a mechanism of ING4-driven TIME modulation. Future investigation in the regulation of specific cytokine and/or MHC class II expression by ING4 may provide insight into the mechanism of CD4 CTL lineage regulation in the TME.

It is notable that our initial hypothesis was based on regulation of NF-kB by ING4 in cancer. Currently it is not clear whether aberrant NF-kB is directly responsible for the CD4 CTL-poor

TIME of *Ing4*-deleted tumors. For example, would aberrant activation of NF-kB in tumors be sufficient to deplete CD4 CTL in *Ing4*-intact TME? By the same token, would inhibition of NF-kB in *Ing4*-deleted tumors restore the CD4 CTL population in the TME? We are currently investigating whether NF-kB impacts CD4 CTL differentiation/activation.

In conclusion, we found the unique tumor immune composition resulting from the absence of *Ing4* in mammary tumors, consisting of decreased CD4 CTLs indicative of an immune-evasive TME, correlating with increased penetrance (fitness for survival) and metastasis. Supportive of the functional contribution of the TIME in ING4-deficient tumor progression, low *GZMB* gene expression together with low *ING4* expression was significantly associated with poor patient survival outcomes in the basal subtype of breast cancer. Taken together, the study results present a novel mechanism of ING4-driven tumor-immune modulation and may offer potential opportunities for therapeutic intervention harnessing CD4 CTLs in ING4-deficient breast cancer.

## Supporting information

**S1 Fig. Uncropped images of Western blots in Fig 2B.**
(PDF)

## Acknowledgments

We thank the Translational Genomics Research Institute for the tumor tissue microarray and histology support. We acknowledge the Flow Cytometry Core facility at the University of Arizona College of Medicine-Phoenix for the resource. We also thank the members of the Kim lab and Hastings lab for helpful discussions.

## Author Contributions

**Conceptualization:** Suwon Kim.

**Data curation:** Emily Tsutsumi, Anne M. Macy, Janine LoBello, Suwon Kim.

**Formal analysis:** Emily Tsutsumi, Anne M. Macy, Karen T. Hastings, Suwon Kim.

**Funding acquisition:** Suwon Kim.

**Investigation:** Suwon Kim.

**Methodology:** Emily Tsutsumi, Anne M. Macy, Janine LoBello, Suwon Kim.

**Resources:** Karen T. Hastings, Suwon Kim.

**Software:** Suwon Kim.

**Supervision:** Karen T. Hastings, Suwon Kim.

**Validation:** Suwon Kim.

**Visualization:** Emily Tsutsumi, Anne M. Macy, Suwon Kim.

**Writing – original draft:** Suwon Kim.

**Writing – review & editing:** Emily Tsutsumi, Anne M. Macy, Karen T. Hastings, Suwon Kim.

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
