## [Decision Letter · Decision Letter 0]

5 Jun 2024

PONE-D-24-17957Tumor immune microenvironment permissive to metastatic progression of ING4-deficient breast cancerPLOS ONE

Dear Dr. Kim,

Thank you for submitting your manuscript to PLOS ONE. After careful consideration, we feel that it has merit but does not fully meet PLOS ONE’s publication criteria as it currently stands. Therefore, we invite you to submit a revised version of the manuscript that addresses the points raised during the review process. Please submit your revised manuscript by Jul 20 2024 11:59PM. If you will need more time than this to complete your revisions, please reply to this message or contact the journal office at plosone@plos.org. Please include the following items when submitting your revised manuscript:A rebuttal letter that responds to each point raised by the academic editor and reviewer(s). You should upload this letter as a separate file labeled 'Response to Reviewers'.A marked-up copy of your manuscript that highlights changes made to the original version. You should upload this as a separate file labeled 'Revised Manuscript with Track Changes'.An unmarked version of your revised paper without tracked changes. You should upload this as a separate file labeled 'Manuscript'.

We look forward to receiving your revised manuscript.

Kind regards,

Yash Gupta, Ph.D.

Academic Editor

PLOS ONE

Journal Requirements:

Additional Editor Comments:

Authors need to add clarity to the methodology and adress reviewers' comments regarding clarity of Gating strategy for flow cytometry in detail. I agree with the reviewer to furnish Western Blot images in the supplementary section. I encourage the authors to elaborate the method section as it is too condensed for readers comfort point of view.

Reviewers' comments:

Reviewer's Responses to Questions

**Comments to the Author**

1. Is the manuscript technically sound, and do the data support the conclusions?

Reviewer #1: Yes

Reviewer #2: Partly

2. Has the statistical analysis been performed appropriately and rigorously? 

Reviewer #1: Yes

Reviewer #2: Yes

3. Have the authors made all data underlying the findings in their manuscript fully available?

Reviewer #1: Yes

Reviewer #2: No

4. Is the manuscript presented in an intelligible fashion and written in standard English?

Reviewer #1: Yes

Reviewer #2: Yes

5. Review Comments to the Author

Reviewer #1: The manuscript submitted by Kim et al, explores the role of ING4 tumor suppressor gene in breast cancer. With a series of mouse experiments supported by analysis of METABRIC gene expression dataset, authors have deciphered the role of ING4 in breast cancer. The manuscript is well written and is easy to comprehend. Interpretation of this paper is well supported by the data. The role of ING4 in cancer in general has been hypothesized in previously published literature, and this work represents a considerable effort to address some of the open questions in this area of research. I recommend this paper to be published in it's current format.

Reviewer #2: In the study entitled "Tumor immune microenvironment permissive to metastatic progression of ING4-

deficient breast cancer" authors have made an excellent effort to understand the mechanism of how ING4-driven tumor-immune modulation creates an immune-evasive tumor microenvironment in breast cancer contributing to metastatic progression and poor patient survival.

But few concerns need attention to meet the publication standard of the journal.

1. Materials and Methods section: please make separate subsections for each and every techniques used for ease of readability and better clarity. For instance, "Tumor Tissue Microarray (TMA) and Immunohistochemistry (IHC)" should be splitted into two separate sub-sections. Same to be followed for rest for the portions in this section where applicable.

2. The IHC images in Figure 1 are low resolution and pixelated. Please replace the same with high resolution clear images.

3. Please explain the 1-3 scoring system of IHC images. Is it done manually or using any specific software to eliminate the bias in scoring.

4. Please include the images of western blots along with protein ladder in supplementary section.

5. Please include the detailed and complete panel of flowcytometry data in supplementary section. the gating strategy is confusing.

6. Please remake figure 5 for better understandability. figure 5B is very confusing.

7. Please include the foldchange readouts of the gene expression data for better clarity and conclusion.

6. PLOS authors have the option to publish the peer review history of their article (what does this mean?). If published, this will include your full peer review and any attached files.

Reviewer #1: **Yes: **Shwetank

Reviewer #2: No

---

## [Author Response · Author response to Decision Letter 0]

18 Jun 2024

We uploaded a letter file responding to the reviewers and editor comments as "Response to Reviewers."

---

## [Editor Report · Decision Letter 1]

21 Jun 2024

Tumor immune microenvironment permissive to metastatic progression of ING4-deficient breast cancer

PONE-D-24-17957R1

Dear Dr. Kim,

We’re pleased to inform you that your manuscript has been judged scientifically suitable for publication and will be formally accepted for publication once it meets all outstanding technical requirements.

Kind regards,

Yash Gupta, Ph.D.

Academic Editor

PLOS ONE
---

## [Editor Report · Acceptance letter]

25 Jun 2024

PONE-D-24-17957R1 

PLOS ONE

Dear Dr. Kim, 

I'm pleased to inform you that your manuscript has been deemed suitable for publication in PLOS ONE. Congratulations! Your manuscript is now being handed over to our production team.

Kind regards, 

on behalf of

Dr. Yash Gupta 

Academic Editor

PLOS ONE